# Assessing target engagement using proteome-wide solvent shift assays

Jonathan G Van Vranken[†], Jiaming Li[†], Dylan C Mitchell, José Navarrete-Perea, Steven P Gygi*

Department of Cell Biology, Harvard Medical School, Boston, United States

**Abstract** Recent advances in mass spectrometry (MS) have enabled quantitative proteomics to become a powerful tool in the field of drug discovery, especially when applied toward proteome-wide target engagement studies. Similar to temperature gradients, increasing concentrations of organic solvents stimulate unfolding and precipitation of the cellular proteome. This property can be influenced by physical association with ligands and other molecules, making individual proteins more or less susceptible to solvent-induced denaturation. Herein, we report the development of proteome-wide solvent shift assays by combining the principles of solvent-induced precipitation (Zhang et al., 2020) with modern quantitative proteomics. Using this approach, we developed solvent proteome profiling (SPP), which is capable of establishing target engagement through analysis of SPP denaturation curves. We readily identified the specific targets of compounds with known mechanisms of action. As a further efficiency boost, we applied the concept of area under the curve analysis to develop solvent proteome integral solubility alteration (solvent-PISA) and demonstrate that this approach can serve as a reliable surrogate for SPP. We propose that by combining SPP with alternative methods, like thermal proteome profiling, it will be possible to increase the absolute number of high-quality melting curves that are attainable by either approach individually, thereby increasing the fraction of the proteome that can be screened for evidence of ligand binding.

## Editor's evaluation

This manuscript will be of broad interest to readers in the field of proteomics and drug discovery. It describes a potentially robust method for the identification of biological targets of small molecules, a substantial hurdle in drug discovery. The experiments described are rigorous, and this manuscript provides a useful template for the broad implementation of this method.

## Introduction

The ability to determine the mechanism of action of a compound of interest starting with an unbiased assessment of its target engagement is a critical aspect of any drug development program. Whether the compound was designed with a specific target in mind or discovered through a phenotypic screen, the ability to define the full range of protein associations is vital in understanding a compound's function and identifying any possible off-target toxicities. In accordance with recent advances in mass spectrometry (MS) and sample multiplexing, proteomics has emerged as a foundational tool for target engagement studies. As such, numerous MS-based methods have been developed to screen the proteome for evidence of ligand binding (*Cousins et al., 2018*; *Daub, 2015*; *Frantzi et al., 2019*; *Moellering and Cravatt, 2012*; *Robers et al., 2020*; *Schürmann et al., 2016*). While each assay has unique advantages and disadvantages, a complete understanding of the mechanism of action of a compound of interest generally relies on the combined efforts of multiple strategies. Therefore, future

*For correspondence:
steven_gygi@hms.harvard.edu

†These authors contributed equally to this work

Competing interest: The authors declare that no competing interests exist.

drug development efforts will benefit from an ever-expanding arsenal of MS-based target engagement approaches.

One powerful MS-based approach for assessing target engagement takes advantage of ligand-induced changes in protein thermal stability. All proteins have a characteristic melting temperature ($T_M$), and this property is often influenced by physical association with other molecules, making a given protein more or less resistant to thermal denaturation. The cellular thermal shift assay (CETSA) and thermal proteome profiling (TPP) technique combine the traditional thermal shift assay with modern quantitative proteomics to generate protein melting curves on a proteome-wide scale (*Jafari et al., 2014*; *Molina et al., 2013*; *Savitski et al., 2014*). As such, individual compounds can be screened against thousands of proteins in a single experiment. While TPP and CETSA have proven to be game-changers in the field of chemo-proteomics and have spawned numerous variations (*Ball et al., 2020*; *Mateus et al., 2020*; *Perrin et al., 2020*; *Saei et al., 2021*), the assays are not without their drawbacks. The reliance on the generation of complete melting curves renders TPP and CETSA inefficient, both in terms of sample preparation and instrument time. Recently, the proteome integral solubility alteration (PISA) assay was proposed to improve the efficiency of proteome-wide thermal shift assays (*Gaetani et al., 2019*). Rather than building complete melting curves, PISA, instead, makes it possible to estimate the area under a protein's melting curve by pooling the soluble fractions of multiple samples heated across a temperature gradient. In the end, target engagement can be determined by simply comparing protein abundance of a compound-treated sample versus a vehicle-treated control. The ability to pool multiple samples vastly increases the throughput of thermal shift assays, while also facilitating complex experimental designs by enabling the simultaneous analysis of multiple compounds, multiple replicates, multiple concentrations, or any combination therein (*Gaetani et al., 2019*; *Li et al., 2020b*). Finally, the ability to represent a ligand binding event as a fold change in measured protein abundance greatly simplifies the data analysis.

Despite the immense value of the two proteome-wide thermal shifts assays described above, alternative and complementary approaches could prove useful both as a means of secondary corroboration and through the expansion of proteomic coverage. Importantly, heat is not the only way to induce unfolding (denaturation) of the proteome. Various agents, including salt, acid, organic solvents, and some chemical denaturants, also disrupt protein folding, leading to aggregation and precipitation (*Chan et al., 1986*; *Meng et al., 2018*; *Shih et al., 1992*; *Wingfield, 2001*). Recently, it was demonstrated that increasing concentrations of an organic solvent mixture composed of 50% acetone, 50% ethanol, and 0.1% acetic acid (AEA) induces precipitation of protein lysates and, more importantly, that this property can be exploited to identify ligand binding events (*Zhang et al., 2020*). This method, termed solvent-induced precipitation (SIP), provided the basic framework for a robust target engagement approach. However, it did not adopt sample multiplexing for quantification of complete melting curves on a proteome-wide scale (throughout this article we will use 'melt' and 'denature,' interchangeably, to generally refer to the process by which a protein transitions from a folded to unfolded state either through the application of heat or exposure to organic solvents). Building on this important study, we sought to develop a proteome-wide solvent shift assay by combining the principles of solvent-induced protein precipitation with modern tandem mass tag (TMT)-based quantitative proteomics. To that end, we defined the solvent-induced denaturation landscape of HCT116 proteome. Furthermore, we established methods that enable the generation and analysis of solvent-induced protein melting curves (solvent proteome profiling [SPP]), improve the overall efficiency of this approach using area under the curve (AUC) analysis (solvent-PISA), and show how each strategy can be used to assess compound target engagement on a proteome-wide scale. Finally, we demonstrate that by combining SPP with TPP or solvent-PISA with thermal-PISA, by extension, we can achieve an expanded view of the cellular proteome.

## Results
### The solvent-induced denaturation landscape of the cellular proteome
With the ultimate goal being the development of a reliable workflow for identifying protein-ligand interactions on a proteome-wide scale, we first sought to explore the impact of increasing concentrations of AEA (50% acetone, 50% ethanol, and 0.1% acetic acid) on the proteome of HCT116 cells. To that end, native HCT116 lysates were treated with increasing concentrations of AEA from 0 to 32.5%

(*Figure 1A*). Protein aggregates were removed from each condition by high-speed centrifugation and the resulting soluble fractions were resolved by SDS-PAGE. While some proteins were insensitive to AEA treatment, the majority of the proteome appeared to denature well between the concentrations tested, reaching a bottom plateau around 25% AEA (*Figure 1—figure supplement 1*).

Next, we wanted to combine the underlying principles of SIP with modern TMT-based quantitative proteomics (*Li et al., 2020a*; *Thompson et al., 2019*). If SIP is to form the foundation for an MS-based target engagement strategy, then three criteria must be met. First, methods to fit protein melting curves to protein abundance data are required. Second, these curves need to be capable of reliably and reproducibly assigning melting concentrations (the concentration of AEA at which a protein is equally distributed between the folded and unfolded state; $C_M$) to individual proteins. Finally, a significant fraction of the proteome needs to be amenable to this method. To address these three objectives, native HCT116 lysates (biological duplicates) were treated with 16 increasing concentrations of AEA from 0 to 21%. An equal volume of each soluble fraction was collected, digested, labeled with one of the 16 TMTpro reagents, and further prepared for LC-MS/MS analysis (*Figure 1A*). In total, this generated two TMTpro 16plexes, one representing each replicate proteome treated with 16 concentrations of AEA. We adapted the R package (TPP) to assign melting curves for TPP data to analyze solvent-induced melting/denaturation curves (see Materials and methods) (*Savitski et al., 2014*). Using TMTpro-based quantitation, sigmoidal protein melting curves were then fit to protein abundance measurements from each %AEA such that a $C_M$ could be assigned to each protein. Similar to thermal melting experiments, a subset of the data is used to normalize protein abundance values at each %AEA (*Figure 1—figure supplement 2*).

In total, we were able to quantify approximately 8500 proteins in each replicate, with greater than 7600 proteins being quantified in both 16plexes (*Figure 1B and C*, *Figure 1—figure supplement 3A*, *Figure 1—source data 1*). Similar to TPP, each curve is assigned three important measures of overall quality—the coefficient of determination ($R^2$), which indicates how well the curve fit the data, the melting concentration ($C_M$), which is given by the concentration at which 50% of the protein is denatured, and the plateau, which is given by the curve's bottom asymptote. Echoing the values used for assigning thermal melting curves, we required that a high-quality solvent melting curve have an $R^2$ value greater than 0.8, a plateau less than 0.3, and a valid slope in order to confidently assign a $C_M$ (*Franken et al., 2015*). Using these filters, we were able to fit high-quality melting curves to 6623 and 6879 proteins in each replicate, respectively, with a total of 5894 curves passing filters in both replicates (*Figure 1B and C*, *Figure 1—figure supplement 3A and B*, *Figure 1—source data 1*). The median $C_M$ in each experiment was 10.2%, with an overall range of approximately 7–21% AEA (*Figure 1D*, *Figure 1—figure supplement 3C*). Individual $C_M$ values correlated well between replicates (*Figure 1E*) and protein melting curves from independent replicates tended to superimpose across a broad range of $C_M$ values (*Figure 1F*). Overall, these data demonstrate that SIP can be employed to reliably and reproducibly assign $C_M$ values to a large fraction (~78%) of the cellular proteome that we were able to detect.

## SPP is capable of target deconvolution

Armed with a workflow capable of generating and interpreting solvent melting curves, we next sought to develop a robust assay for assessing target engagement. To that end, we developed SPP, an MS-based sample multiplexing approach for uncovering protein-ligand interactions on a proteome-wide scale (*Figure 2A*, *Figure 2—figure supplement 1*). To test this workflow, we selected SCIO-469, an inhibitor of p38 MAP kinases, for analysis by SPP (*Dominguez et al., 2005*; *Hideshima et al., 2004*). HCT116 lysates were treated with 100 µM SCIO-469 or vehicle (DMSO). Following a brief incubation, each sample was further divided into eight aliquots and treated with increasing concentrations of AEA from 0 to 21% (0, 3, 6, 9, 12, 15, 18, and 21%). Protein aggregates were removed by high-speed centrifugation and an equal volume of each soluble fraction was collected and prepared for downstream analysis.

Moving from 16 concentrations to 8 allowed us to combine the vehicle- and compound-treated samples into a single TMTpro 16plex, thereby assuring all detected proteins will be quantified in both the control and treated groups (*Zinn et al., 2021*). In order to confirm that this transition did not have a negative impact on our ability to fit solvent melting curves, we first inspected the overall quality of the dataset. In total, we were able to quantify 9118 and 8847 proteins in each replicate, respectively

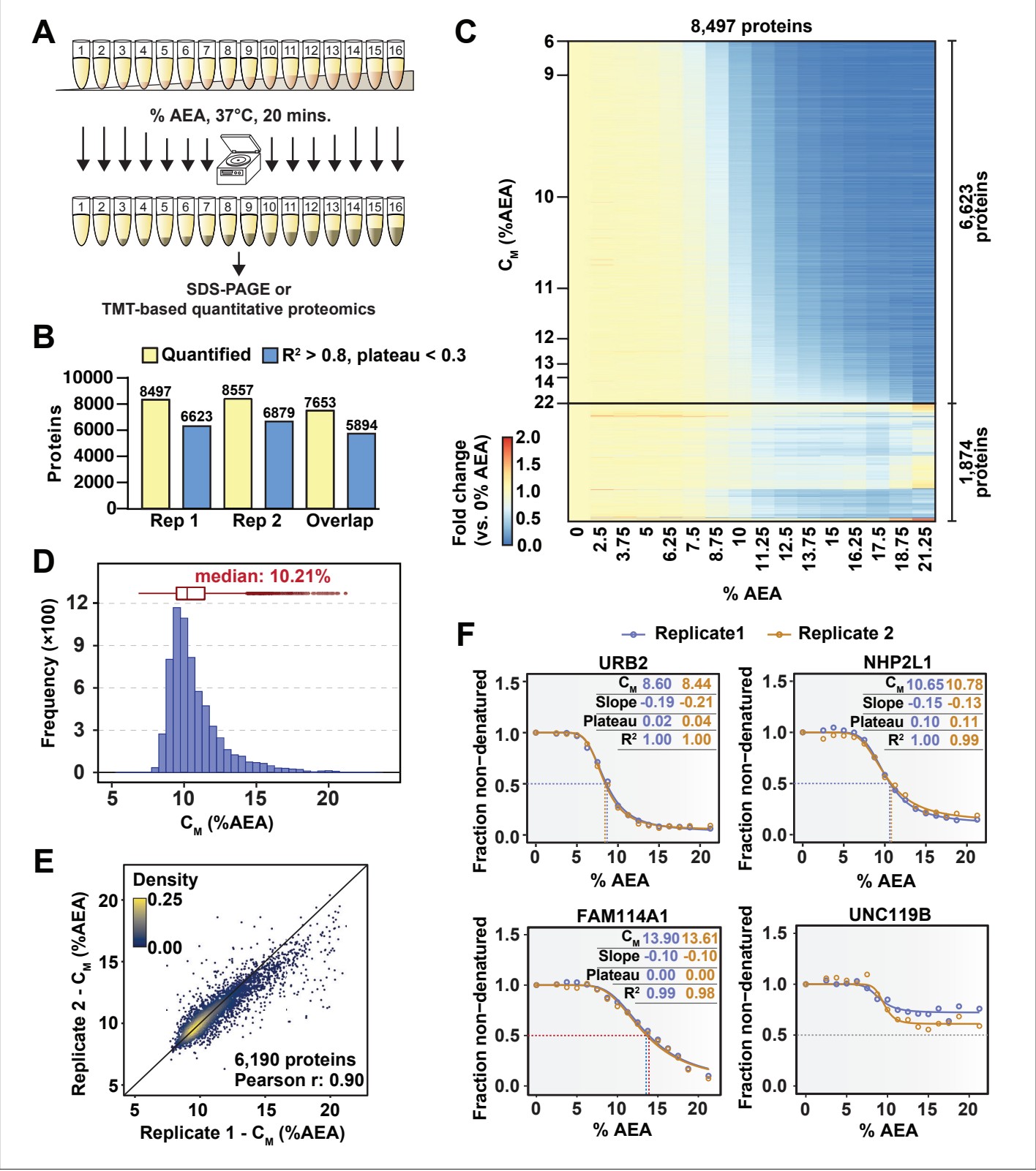

**Figure 1.** Solvent profiling of the HCT116 proteome. (**A**) Schematic diagram of solvent-induced precipitation. (**B**) Count of quantified proteins in each replicate and those to which sigmoidal curves were fit well ($R^2 > 0.8$ and plateau < 0.3). Each replicate is a single TMTpro 16plex experiment. (**C**) Heatmap representation of all proteins quantified in replicate 1. For each protein, its relative abundance (fold change) at the indicated %AEA compared to 0% AEA is presented. The proteins for which high-quality curves ($R^2 > 0.8$ and plateau < 0.3) could be obtained (6623) are separated from those for

*Figure 1 continued on next page*

*Figure 1 continued*

which curves with reduced quality fits were returned (1874). Proteins are sorted by $C_M$. (**D**) $C_M$ distribution for replicate 1. Proteins to which sigmoidal curves were fit well were included (6623 proteins, $R^2 > 0.8$ and plateau < 0.3). (**E**) Reproducibility of $C_M$ measures between replicates. A high correlation (Pearson correlation – 0.9) was achieved. Proteins that showed high-quality curves ($R^2 > 0.8$ and plateau < 0.3) in at least one replicate were included. (**F**) Examples of solvent melting curves for URB2, NHP2L1, FAM114A1, and UNC119B from replicate 1 (blue) and replicate 2 (orange). Inset in each panel reports $C_M$, slope, plateau, and $R^2$ for each curve. Curves were selected to highlight a range of $C_M$ values.

The online version of this article includes the following figure supplement(s) for figure 1:

**Source data 1.** Protein quantifications in the solvent-induced precipitation assay with 16 AEA concentrations (N = 2).

**Figure supplement 1.** Solvent-induced denaturation of the HCT116 proteome.

**Figure supplement 2.** Solvent profiling of the HCT116 proteome.

**Figure supplement 3.** Solvent profiling of the HCT116 proteome.

(*Figure 2—figure supplement 2A–D*, *Figure 2—source data 1*). After curve fitting, $C_M$ values could be assigned to more than 6500 proteins in each group (~75% of quantified proteins), a value in line with what was previously observed when using 16 concentrations (*Figure 1*). Using these eight-point melting curves, we next assigned a $C_M$ to each protein in the control and treated groups and found a median $C_M$ of approximately 12% across all four groups (*Figure 2—figure supplement 2E–H*). This is in contrast to the median $C_M$ of ~10%, which was observed in *Figure 1*. We believe that this disparity stems from day-to-day variation in the preparation of AEA stock solutions. Nonetheless, while there appears to be some amount of deviation between experiments, replicate data, produced using the identical AEA stocks, generate $C_M$ values that are highly reproducible (*Figure 2—figure supplement 2I and J*). Overall, these data demonstrate that high-quality solvent melting curves can be generated from just eight concentrations of AEA.

We next directed our attention to the known targets of SCIO-469. Appropriately, all three of the p38 MAP kinases that we detected—MAPK14, MAPK13, and MAPK12—exhibited an evident shift in $C_M$ in response to SCIO-469 treatment (*Figure 2B*, *Figure 2—figure supplement 3*, *Figure 2—source data 1*). Additionally, the compound-dependent stabilization of p38 was independently corroborated by western blot, thereby validating the MS-based quantitation (*Figure 2E*).

Satisfied with the outcome of our initial SPP trial, we applied this workflow to two additional compounds—Alisertib and MK2206—which engage and inhibit Aurora kinase A (AURKA) and AKT, respectively (*Bavetsias and Linardopoulos, 2015*; *Hirai et al., 2010*). Native HCT116 lysates were treated with 25 µM Alisertib, 25 µM MK2206, or vehicle (DMSO) and subjected to SPP. Alisertib is a potent and specific inhibitor of AURKA, which it inhibits with at least two orders of magnitude greater potency than Aurora kinase B (AURKB) (*Bavetsias and Linardopoulos, 2015*). Consistent with this disparity, Alisertib induces an evident shift in $C_M$ for AURKA, while AURKB did not experience such a shift in response to the compound (*Figure 2C and E*, *Figure 2—source data 2*). MK2206 is a pan-AKT inhibitor and, as such, causes a significant shift in both AKT1 and AKT2 (*Figure 2D and E*, *Figure 2—source data 3*). Overall, these data further reinforce that SPP can be used to reliably assess compound target engagement on a proteome-wide scale.

## Solvent-PISA increases the efficiency of SPP

Similar to TPP, SPP depends on building and analyzing full protein melting curves and, therefore, suffers from similar drawbacks as its predecessor. The PISA technique was developed to address these issues and increase the efficiency of proteome-wide thermal shift assays. Rather than relying on analysis of full melting curves, thermal-PISA, instead, approximates the area under a protein melting curve by pooling samples that had been heated across a temperature gradient (*Gaetani et al., 2019*). We applied this strategy to SPP to develop solvent-PISA (*Figure 3A*). Rather than labeling each soluble fraction individually, as in SPP, an equal volume from each sample is pooled together. This pooled sample, which approximates the area under the solvent melting curve for all proteins quantified, can then be labeled with a single TMTpro reagent. Importantly, comparing a single compound to a vehicle-treated control using solvent-PISA requires just two TMTpro channels (each channel represents thousands of melting curves from a single sample), whereas the analogous SPP experiment would use the full 16 channels. Thus, solvent-PISA can support more complex experimental design that allows for the simultaneous analysis of multiple replicates, multiple concentrations, or multiple compounds

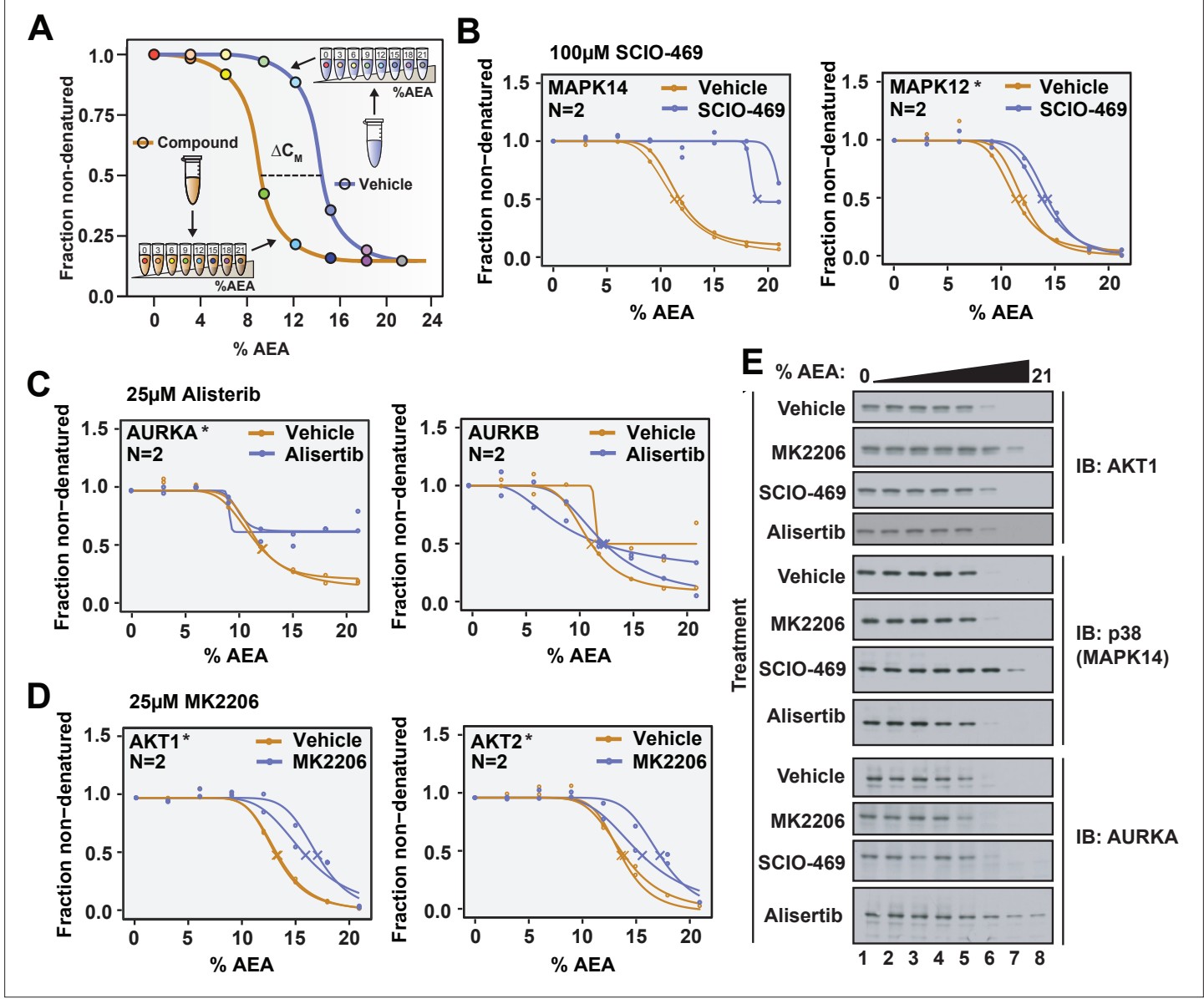

**Figure 2.** Solvent proteome profiling (SPP) can be used to determine compound target engagement. (**A**) Schematic diagram summarizing the SPP workflow. (**B**) Native HCT116 lysates were treated with 100 μM SCIO-469. SPP melting curves for MAPK14 and MAPK12—known target proteins of SCIO-469—are displayed from two biological replicates. (**C**) Native HCT116 lysates were treated with 25 μM Alisertib. SPP melting curves for AURKA and AURKB are displayed for two biological replicates. (**D**) Native HCT116 lysates were treated with 25 μM MK2206. SPP melting curves for AKT1 and AKT2 are displayed for two biological replicates. (**E**) Soluble fractions from SPP were separated by SDS-PAGE and immunoblotted with the indicated antibodies. Asterisks indicate statistically significant hits (see Materials and methods). Crosses indicate melting points when available.

The online version of this article includes the following source data and figure supplement(s) for figure 2:

**Source data 1.** Solvent proteome profiling (SPP) protein quantifications following treatment of HCT116 lysates with 100 μM SCIO-469 or vehicle.

**Source data 2.** Solvent proteome profiling (SPP) protein quantifications following treatment of HCT116 lysates with 25 μM Alisertib or vehicle.

**Source data 3.** Solvent proteome profiling (SPP) protein quantifications following treatment of HCT116 lysates with 25 μM MK-2206 or vehicle.

**Source data 4.** Unedited film scans of Akt western blots (Cell Signaling Technology, 4691). (**A**) Vehicle and MK2206 treatment. (**B**) SCIO-469 treatment. (**C**) Alisertib (MLN8237) treatment.

**Source data 5.** Unedited film scans of p38 western blots (Cell Signaling Technology, 9212). (**A**) Vehicle and MK2206 treatment. (**B**) SCIO-469 treatment. (**C**) Alisertib (MLN8237) treatment.

**Source data 6.** Unedited film scans of Aurora kinase A western blots (Cell Signaling Technology, 4718). (**A**) Vehicle treatment. (**B**) SCIO-469 and MK2206 treatment. (**C**) Alisertib (MLN8237) treatment. Blue boxes indicate the blot used to generate final figure.

*Figure 2 continued on next page*

*Figure 2 continued*

**Figure supplement 1.** Schematic diagram of solvent proteome profiling (SPP).

**Figure supplement 2.** Solvent proteome profiling (SPP) of the HCT116 proteome.

**Figure supplement 3.** Solvent proteome profiling (SPP) can be used to determine compound target engagement.

in a single TMTpro group. In the end, ligand binding can be read out as a simple increase or decrease in protein abundance in the compound-treated sample versus the vehicle-treated control often determined by a simple t-test (*Figure 3A*).

In order to illustrate the utility of solvent-PISA and underscore its increased capacity for sample multiplexing, we again treated native HCT116 lysates (N = 4) with 100 μM SCIO-469 (to mirror SPP data) or vehicle (DMSO). The samples were exposed to eight increasing concentrations of AEA from 9 to 19%. After centrifugation to remove aggregates, an equal volume of each soluble fraction was pooled, labeled with TMTpro reagents, and prepared for LC-MS/MS analysis. In total, we quantified ~7200 proteins with two or more peptides and found that the known targets of SCIO-469, including MAPK14 and MAPK12, emerged as significant hits (*Figure 3B*, *Figure 3—source data 1*). In addition to these known targets, solvent-PISA also highlighted additional putative targets (*Figure 3B*). One of these potential targets, MAPK9/JNK2, also scored in the full-curve SPP experiment (*Figure 3—figure supplement 1A*). Therefore, we sought to confirm this result using an in vitro kinase assay and found that SCIO-469 could inhibit MAPK9/JNK2 (*Figure 3—figure supplement 1B*). Interestingly, other putative targets identified in the PISA experiment, including CDK19, did not score in the full-curve SPP experiment. Upon closer inspection, it was discovered that CDK19 appeared to shift $C_M$ in the presence of SCIO-469. However, it was only found in a single SPP replicate and was therefore ignored (*Figure 3—figure supplement 1C*). Overall, these data demonstrate that solvent-PISA is an adequate surrogate for SPP, capable of identifying protein-ligand interactions using AUC analysis.

Next, we wanted to compare solvent-PISA to thermal-PISA. To that end, we included a second compound, in addition to SCIO-469, and selected vorinostat (*Marks and Breslow, 2007*), a histone deacetylase (HDAC) inhibitor that, like SCIO-469, we had previously profiled with thermal-PISA and, in the context of solvent-PISA, causes a significant shift of its known targets (*Figure 3C*). In order to compare the two approaches (which used a varying number of replicates), we first calculated a %CV for each group and found that >97.5% of the data had CVs under 15% across all groups (*Figure 3—figure supplement 2A–D*, *Figure 3—source data 1*, *Figure 3—source data 2*). We removed proteins with CVs greater than 15% across replicates and ranked the remaining proteins by their fold changes. In general, a similar set of proteins topped both lists in response to each compound (*Figure 3—figure supplement 3A and B*, *Figure 3—source data 1*, *Figure 3—source data 2*). For example, MAPK14, MAPK12, and MAPK9 had altered AUC values in both assays in response to SCIO-469 (*Figure 3*, *Figure 3—figure supplement 3A and B*). Following treatment with vorinostat, five proteins, including two known targets HDAC1 and HDAC2, were among the largest fold changes in both datasets. We also observed a large number of proteins with a negative log2 fold change using both solvent- and thermal-PISA. While it is possible that some of these changes reflect ligand binding, it is unlikely that all of these proteins engage vorinostat and more probable that these changes stem from an artifact associated with the chemical properties of this particular compound as a similar effect has not been observed for any other compounds tested (*Figure 3E*, *Figure 3—figure supplement 3C and D*). Despite the evident similarities of the two approaches, they do not appear to be completely redundant. In fact, there are numerous proteins that emerged from only a single approach. MAPK8 and CDK19, for example, are altered exclusively in solvent-PISA upon SCIO-469 treatment, while HDAC6 is altered exclusively in thermal-PISA following vorinostat treatment (*Figure 3D and E*). Overall, these data suggest that while these two approaches are capable of generating a similar set of putative targets, these lists are not completely identical. Therefore, SPP and TPP appear to be complimentary not only because they can provide independent corroboration but because one method could potentially identify a target that the other might miss.

Having previously performed SPP on both Alisertib and MK2206, we next analyzed these two compounds by solvent-PISA. This provided an opportunity to highlight the increased multiplexing capacity of this approach by simultaneously analyzing multiple concentrations in a single experiment. HCT116 lysates (N = 4) were treated with three increasing concentrations (5 μM, 25 μM, and 125 μM)

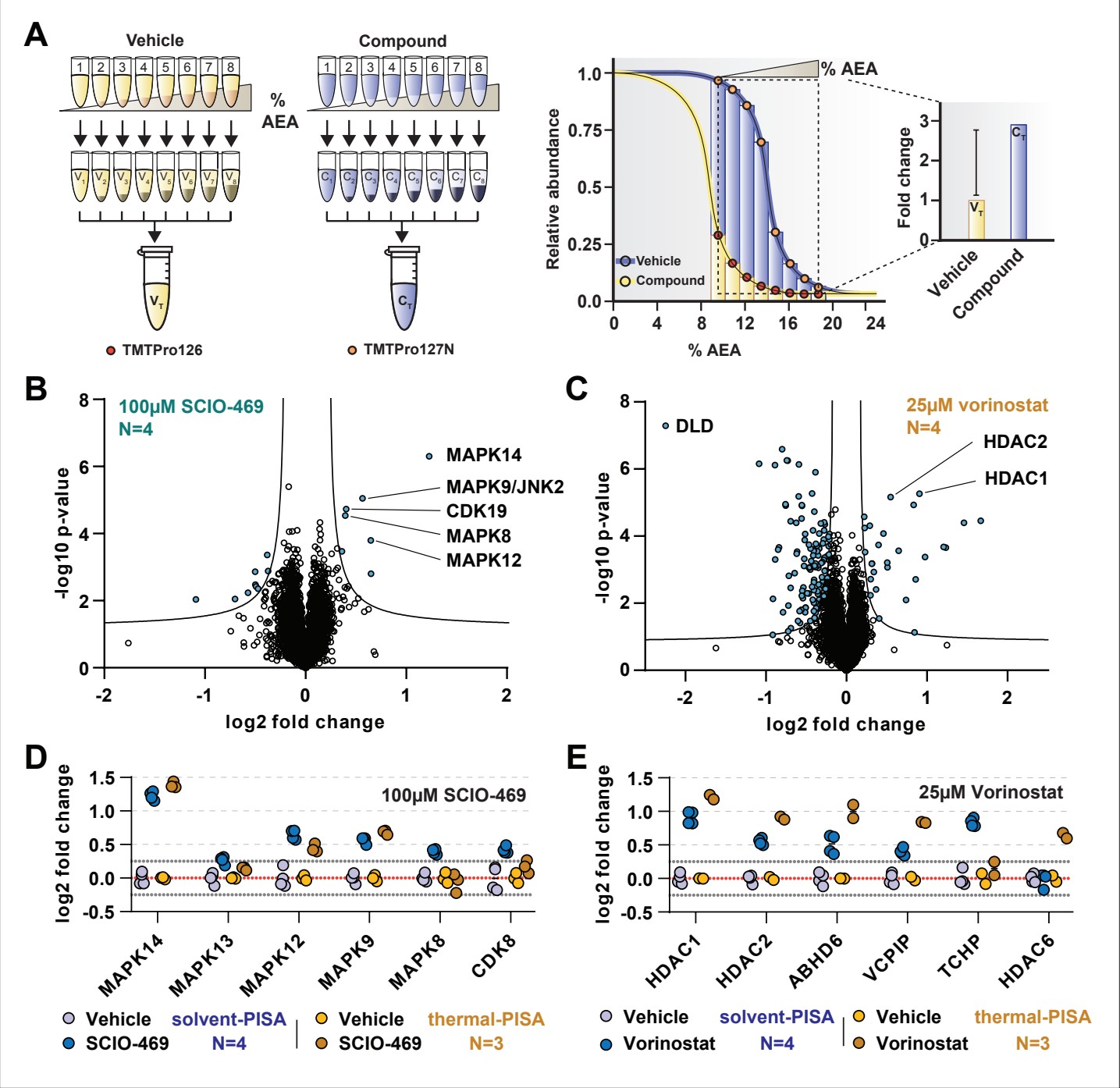

**Figure 3.** Solvent-PISA can resolve compound target engagement with increased efficiency. (**A**) Native cell lysates are prepared and divided into multiple aliquots. Half the aliquots are treated with drug and the other half reserved for a vehicle-treated control. Samples are incubated for 15 min at room temperature. Compound- and vehicle-treated samples are further divided into eight aliquots and treated with increasing concentrations of AEA (9, 10.375, 11.75, 13.125, 14.5, 15.875, 17.25, and 18.625%). AEA-treated samples are incubated at 37°C for 20 min at which point they are subjected to centrifugation to separate the soluble and insoluble fractions. Soluble fractions are collected and pooled in equal volumes before being reduced, alkylated, and digested. The resulting peptides are labeled with TMTpro 16plex reagents, pooled, and fractionated before being analyzed by LC-MS/MS. Target engagement is determined by an increase or decrease in protein content compared to the vehicle-treated control. (**B, C**) HCT116 lysates were treated with 100 µM SCIO-469 (**B**) or 25 µM vorinostat (**C**) and analyzed by solvent-PISA. Data are presented as a volcano plot to highlight significant changes in abundance. Significant changes were determined using a permutation-based false discovery rate (FDR) (FDR – 0.05, S0 – 0.1) and are indicated with blue dots. (**D, E**) HCT116 lysates were treated with 100 µM SCIO-469 (**D**) or 25 µM vorinostat (**E**) and analyzed by solvent-PISA (blue shades) or thermal-PISA (orange shades). Individual log2 fold change values (in reference to the vehicle mean) are plotted for several proteins. Dotted

*Figure 3 continued on next page*

*Figure 3 continued*

lines at y = 0.25 and y = −0.25 are included to denote fold change values of ~20%.

The online version of this article includes the following figure supplement(s) for figure 3:

**Source data 1.** Solvent-PISA and thermal-PISA protein quantifications following treatment of HCT116 lysates with 100 µM SCIO-469 or vehicle (N = 4).

**Source data 2.** Solvent-PISA and thermal-PISA protein quantifications following treatment of HCT116 lysates with 25 µM vorinostat or vehicle (N = 4).

**Figure supplement 1.** Solvent-PISA can resolve compound target engagement with increased efficiency.

**Figure supplement 2.** Solvent-PISA can resolve compound target engagement with increased efficiency.

**Figure supplement 3.** Solvent-PISA can resolve compound target engagement with increased efficiency.

of each compound and analyzed by solvent-PISA (*Figure 4—source data 1*, *Figure 4—source data 2*). Consistent with previous SPP data, the known targets of each compound emerged as significant hits following 25 µM treatments (*Figure 4A*, *Figure 4—figure supplement 1A-C* ). AURKA reached a maximum log2 fold change at the lowest dose of Alisertib assayed and held steady upon treatment with increasing doses (*Figure 4—figure supplement 1D*). Treatment with increasing concentrations of MK2206, on the other hand, resulted in a dose-dependent increase in log2 fold change for its targets—AKT1 and AKT2 (*Figure 4B*). Overall, these data further reinforce the utility of solvent-PISA and underscore its benefits compared to SPP.

Previously, we demonstrated that, in the context of thermal-PISA, the selection of temperature range ultimately impacts the fold change values, with higher and smaller temperature windows generally resulting in larger fold changes (*Li et al., 2020b*). In this study, we have focused on a range of 9–19% AEA for all solvent-PISA experiments thus far. This range encompasses the back half of most solvent-induced protein melting curves, a concept that we found to be advantageous for thermal-PISA experiments. Nonetheless, we wanted to understand how the selection of AEA concentrations impacts the final fold changes that are measured in a solvent-PISA experiment. Therefore, we exposed HCT116 lysates (N = 2) treated with 25 µM MK2206 to four ranges of AEA concentrations, which we will refer to as windows (*Figure 4C*, *Figure 4—source data 3*). The first window covered the entire melting curve from 0 to 21% AEA. The remaining three windows covered much smaller and higher ranges and were all localized to the back half of most melting curves (*Figure 4C*). Echoing what was seen previously for thermal-PISA, the selection of %AEA range had a profound impact on the ultimate log2 fold changes that were measured, with progressively later and smaller windows resulting in larger fold change values (*Figure 4D*). Focusing on AKT1 and AKT2, using the full window of 0–21% AEA resulted in a measured log2 fold change of approximately 0.3 and 0.2, respectively. Utilizing the remaining three windows, all resulted in much larger changes, with AKT1 exhibiting log2 fold changes of ~0.6, ~0.75, and ~1.6 for windows 2 (9–19% AEA), 3 (11–19% AEA), and 4 (14–19%), respectively, while AKT2 followed a similar trend. Having previously performed SPP on MK2206-treated lysates, we took the opportunity to calculate the expected log2 fold change for each %AEA window based on the SPP melting curves (*Figure 2D*) and found that the calculated (*Figure 4D*) and measured (*Figure 4E*) log2 fold change values were quite similar. Overall, these data demonstrate that the selection of AEA concentrations meaningfully impacts the final dataset such that the chosen range should be carefully considered.

While it is likely that AKT1 and AKT2 would have been identified using any of the windows, we found at least one example of a putative target that only experienced a significant fold change in a single solvent-PISA window. Indeed, CHEK1 does not appear to change substantially in any of the first three windows but exhibits an average log2 fold change of −0.7 upon utilization of window 4 (*Figure 4D*). Consistent with our previous observations, these values also correlate well with the calculated values based on the CHEK1 solvent melting curves (*Figure 4E*). Upon closer inspection of SPP data, it is clear that CHEK1 has a high $C_M$ and treatment with 25 µM MK2206 resulted in a slight but reproducible $\Delta C_M$ (*Figure 4—figure supplement 2A*). This likely explains why CHEK1 only emerged as a putative target when utilizing the smallest solvent-PISA window. Finally, we found that treatment with higher doses of MK2206 (125 µM) in an independent experiment caused a significant decrease (log2 fold change of −0.52) in CHEK1, even when utilizing the window of 9–19% AEA (*Figure 4—figure supplement 2B and C*). While these data imply that CHEK1 might be a target of MK2206, they are far from definitive and would require further investigation.

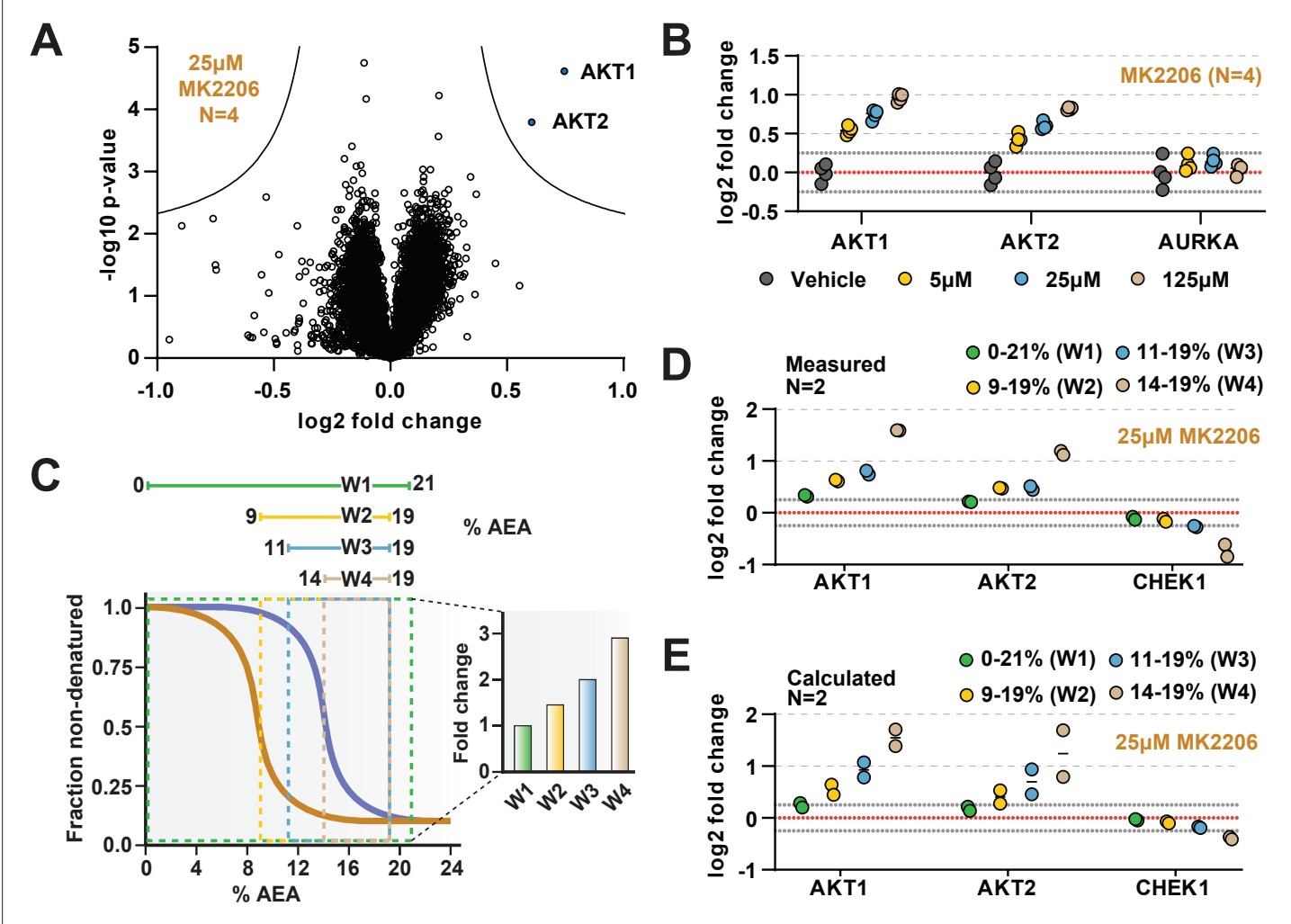

**Figure 4.** The range of AEA concentrations used in a solvent-PISA experiment impacts the ultimate fold change measurements.
 (**A**) HCT116 lysates were treated with 25 μM MK2206 and analyzed by solvent-PISA. Data are presented as a volcano plot to highlight significant changes in abundance. Significant changes were determined using a permutation-based false discovery rate (FDR) (FDR – 0.05, S0 – 0.1) and are indicated with blue dots. (**B**) HCT116 lysates were treated with 5 μM, 25 μM, or 125 μM MK2206 and analyzed by solvent-PISA. Individual log2 fold change values are plotted for several proteins at each concentration. Dotted lines at y = 0.25 and y = −0.25 are included to highlight minimum changes of ~20%. (**C**) Schematic diagram of solvent proteome profiling (SPP) melting curves indicating the range of %AEA used for each window. (**D**) Individual log2 fold change values are plotted for several proteins measured using different windows. (**E**) Expected log2 fold change values for several proteins were calculated based on SPP melting curves determined by SPP for each window (calculated from *Figure 2B*). Individual abundances for the compound-treated samples and vehicle-treated controls at each %AEA within a given range were summed. Considering the 0–21% window (window 1), for example, we simply summed all eight abundance measurements from the eight AEA concentrations used to generate the SPP data. For the 9–19% window (window 2), we summed only the SPP abundances between 9 and 19%, and so on.

The online version of this article includes the following figure supplement(s) for figure 4:

**Source data 1.** Solvent-PISA protein quantifications following treatment of HCT116 lysates with 25 μM Alisertib or vehicle (N = 4).

**Source data 2.** Solvent-PISA protein quantifications following treatment of HCT116 lysates with 25 μM MK2206 or vehicle (N = 4).

**Source data 3.** Solvent-PISA protein quantifications following treatment of HCT116 lysates with 25 μM MK2206 or vehicle across four solvent-PISA windows (see Materials and methods) (N = 2).

**Figure supplement 1.** The range of AEA concentrations used in a solvent-PISA experiment impacts the ultimate fold change measurements.

**Figure supplement 2.** The range of AEA concentrations used in a solvent-PISA experiment impacts the ultimate fold change measurements.

## Combining solvent shift assays with thermal shift assays increases proteome coverage

So far, we have described two new strategies for assessing target engagement on a proteome-wide scale. While only SPP involves building complete protein melting curves, solvent-PISA still relies on them to a large extent. That is, if a high-quality melting curve cannot be fit to a given protein, the protein is not amenable to analysis by SPP and very likely not by solvent-PISA, either. On the basis of previously published TPP data and the SPP data presented here, it appears that high-quality curves cannot be fit to as much as 30% of quantified proteins (*Figure 5A*, *Figure 5—figure supplement 1A*; *Jarzab et al., 2020*). We hypothesized that by combining SPP with TPP or, indeed, solvent-PISA with thermal-PISA, it might be possible to expand the overall coverage of the proteome, allowing access to more proteins than either approach, individually by maximizing the absolute number of high-quality curves. To test this hypothesis, we generated native HCT116 lysates (biological duplicates) and performed both SPP and TPP. SPP was performed using eight different concentrations of AEA, while TPP experiments were performed using eight temperatures. Biological duplicates were combined into a single TMTpro 16plex to maximize the overlap between replicates. Each approach resulted in the quantification of ~8500 proteins. However, 25–30% of each dataset were lost due to poor curve quality, meaning that only approximately 6500 proteins could be assigned a $C_M$ or $T_M$ (*Figure 5A and B*, *Figure 5—figure supplement 1A*, *Figure 5—source data 1*). A total of 7667 proteins were quantified using both SPP and TPP. Using $R^2 > 0.8$ and plateau < 0.3 as criteria, high-quality curves could be fit to approximately 5000 proteins in both approaches, while over 1000 proteins could not be assigned a good curve using either SPP or TPP (*Figure 5C*, *Figure 5—figure supplement 1B and C*). Consistent with our hypothesis, there were over 620 proteins that produced high-quality curves in SPP but not TPP and over 850 proteins that produced high-quality curves in TPP but not SPP. Therefore, by combining both SPP and TPP, we were able to fit high-quality curves to a total of over 6400 proteins, which is more than either approach, independently (*Figure 5C*, *Figure 5—figure supplement 1C*, *Figure 5—source data 1*). While these numbers will, of course, vary based on experimental conditions such as proteome extraction method and AEA/temperature range, we expect that the basic trend will hold true. Thus, we conclude that combining the two methods can expand total proteomic coverage and allow a molecule of interest to be screened against a greater fraction of the total cellular proteome.

## Discussion

Assessing target engagement on a proteome scale relies on MS-based approaches that can help determine mechanism of action and identify potential off-target toxicities. Due to the complex nature of these studies, an ever-expanding repertoire of available methodologies is critical when interrogating the function of compounds with potential therapeutic value. Using the principles of SIP as a foundation (*Zhang et al., 2020*), we have developed proteome-wide solvent shift assays in the form of SPP and solvent-PISA, which represent a significant advancement compared to the previous iteration. First, we demonstrate that increasing concentrations of AEA are capable of inducing proteome denaturation that is reminiscent of thermal shift assays (*Figure 1*). Furthermore, we present methods to fit protein abundance data with sigmoidal melting curves, which facilitates the assignment of $C_M$ values to a significant fraction of the cellular proteome (*Figure 1*). Based on these initial findings, we then developed SPP, which utilizes full melting curves and $\Delta C_M$ measurements to establish compound target engagement. In order to highlight the utility of this approach, we assayed several compounds in HCT116 lysates, pinpointing compound-dependent shifts in $C_M$ for the known targets (*Figure 2*). In order to improve the efficiency of proteome-wide solvent shift assays, we applied the concept of AUC analysis to SPP to develop solvent-PISA, which relies on pooling multiple samples treated with increasing concentrations of AEA. Using this approach, we were able to assign target engagement based on a simple comparison of protein abundances between compound- and vehicle-treated samples and highlight the ability of solvent-PISA to increase the multiplexing potential of these assays (*Figures 3 and 4*). Finally, we find that by combining solvent and thermal shift assays it is possible to cover a greater fraction of the cellular proteome than either approach individually, thereby maximizing the absolute number of high-quality curves (*Figure 5*).

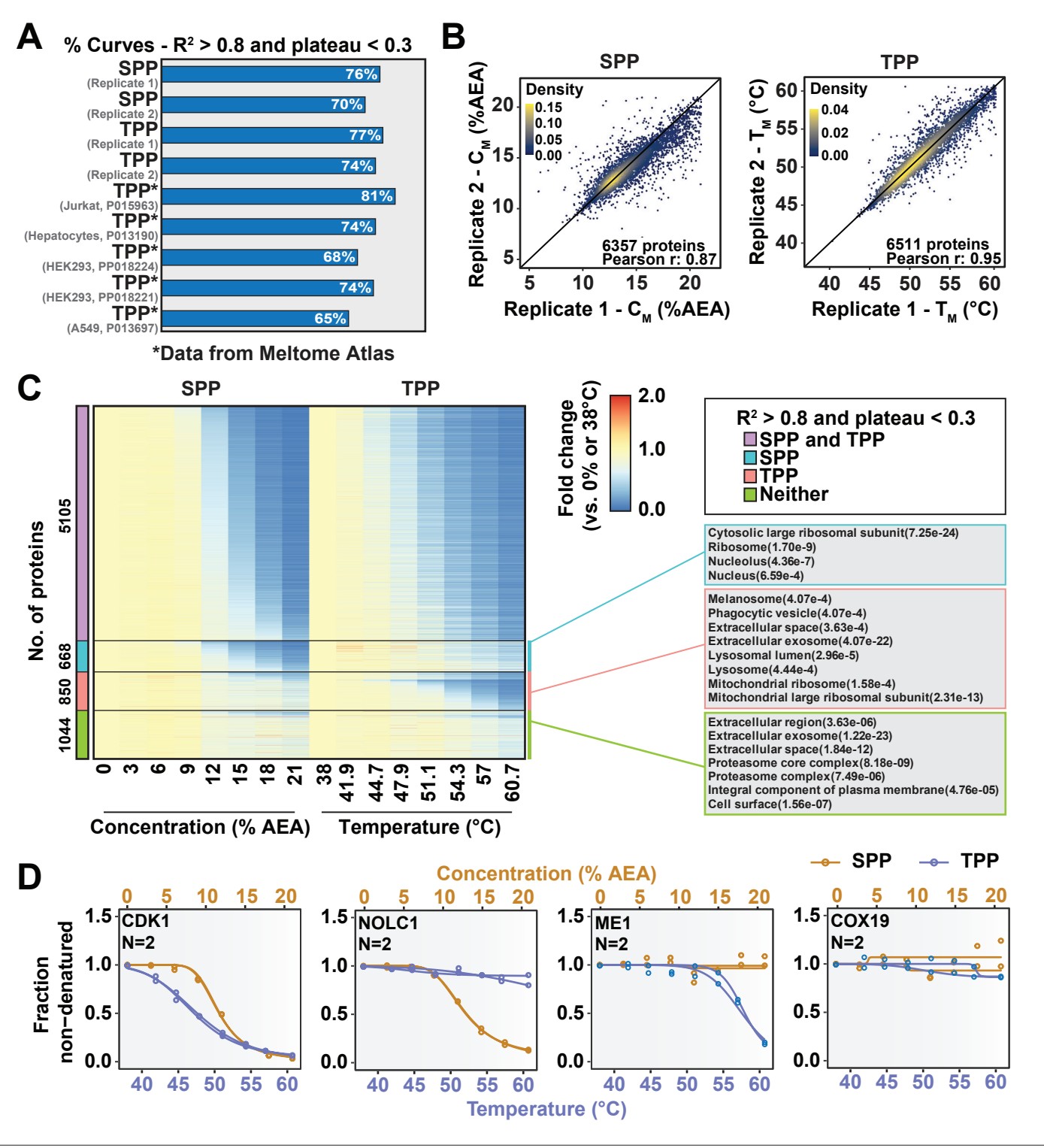

**Figure 5.** Combining solvent proteome profiling (SPP) with thermal proteome profiling (TPP) to maximize proteome coverage. (**A**) Fractions of high-quality curves ($R^2 > 0.8$ and plateau < 0.3) in SPP and TPP datasets. * indicates datasets taken from the Meltome Atlas (***Jarzab et al., 2020***). A similar fraction (~70%) of high-quality curves were obtained in SPP and TPP assays. (**B**) Reproducibility of SPP $C_M$ (left panel) and TPP $T_M$ (right panel) values between replicates. SPP and TPP experiments showed equivalent reproducibility between replicates. Proteins that show high-quality curves ($R^2 > 0.8$ and plateau < 0.3) in at least one replicate were included. (**C**) Heatmap representation of all proteins quantified in both SPP and TPP (replicate 1). For each protein, its relative abundance (fold change) at the indicated %AEA compared to 0% AEA (SPP, left) or at the indicated temperature compared to 38°C

*Figure 5 continued on next page*

*Figure 5 continued*

(TPP, right) is presented. The panel on the right indicates Gene Ontology entries that are enriched in each indicated group. Proteins that were quantified in both SPP and TPP were used as the background. (**D**) Selected SPP (orange) and TPP (blue) denaturation/melting curves highlighting proteins showing high-quality curves ($R^2 > 0.8$ and plateau < 0.3) in both approaches, in SPP alone (second), in TPP alone (third), or neither (fourth).

The online version of this article includes the following figure supplement(s) for figure 5:

**Source data 1.** Solvent proteome profiling (SPP) and thermal proteome profiling (TPP) protein quantifications (eight AEA concentrations or eight temperatures) (N = 2).

**Figure supplement 1.** Combining SPP with TPP to maximize proteome coverage.

The solvent shift assays introduced here echo many important features of proteome-wide thermal shift assays—TPP and thermal-PISA (*Gaetani et al., 2019*; *Savitski et al., 2014*). Upon analysis of SPP data, we found that it was possible to assign high-quality melting curves and $C_M$ values to a similar but nonredundant proportion of the total proteome compared to TPP. Furthermore, like TPP, SPP can encounter challenges in fitting and analyzing melting curves. For example, MAPK14 and AURKA exhibited obvious solvent shifts in response to SCIO-469 and Alisertib (*Figure 2B and C*), respectively; however, the $C_M$ shifted to the point where they were no longer amenable to statistical analysis via the modified TPP package because the treated group failed curve quality filters. While we used the TPP package here to generate the denaturation landscape of HCT116 proteome and benchmark SPP against TPP, it is important to note that alternative statistical methods exist that may be able to handle ill-shaped curves (*Childs et al., 2019*). In terms of solvent-PISA, we were again able to cover a similar fraction of the proteome and demonstrate that, in comparison to thermal-PISA, it is generally possible to identify a similar set of targets in response to compound treatment. Solvent-PISA can also overcome poor curve quality to some extent as both MAPK14 and AURKA were recovered as statistically significant hits in solvent-PISA (*Figure 3B*, *Figure 4—figure supplement 1B*). Furthermore, we found that varying the range of AEA concentrations (solvent windows) used in a solvent-PISA experiment impacts the magnitude of the ultimate fold change measurements, a phenomenon that is also observed for thermal-PISA and the selection of temperature ranges (thermal windows). So, while proteome-wide solvent and thermal shift assays behave comparably in lysate-based experiments, SPP and solvent-PISA are not amenable to cell-based approaches, like CETSA (*Jafari et al., 2014*; *Molina et al., 2013*). This is because, unlike heat, the plasma membrane will likely impede the ability of AEA to access with the cellular proteome of living cells, thereby preventing solvent-induced protein precipitation. In the end, this prevents SPP and solvent-PISA from being applied to certain applications (*Becher et al., 2018*; *Dai et al., 2018*), but it does not impact the ability of these assays to define direct ligand binding in lysate-based experiments.

SPP and TPP represent two independent approaches that ultimately depend on the analysis of full protein melting or denaturation curves. Therefore, the ability to maximize the collection of high-quality curves will increase proteomic coverage and make these approaches more powerful. In this study, we propose that by utilizing alternative precipitation methods it is possible to increase the absolute number of high-quality curves. Importantly, these are not the only strategies worth considering. Utilizing additional means of proteome denaturation like salt, acid, additional solvents, chemical denaturants, and others could allow access to a greater number of high-quality curves (*Meng et al., 2018*; *Savitski et al., 2014*; *Zhang et al., 2020*). Additionally, altering proteome extraction conditions could also increase the proteomic coverage. In this report, we generated all native cell lysates using 0.5% NP-40. However, these assays are amenable to a myriad of extraction strategies (*Reinhard et al., 2015*; *Savitski et al., 2014*; *Sridharan et al., 2019*). As such, utilizing multiple detergents at varying concentrations or no detergents at all could provide conditions that favor melting of certain proteins but not others. While AUC approaches (PISA) do not explicitly rely on building curves to determine engagement, these curves are still important, and while it remains formally possible that PISA could overcome poor curve quality, more high-quality curves will still ultimately prove beneficial. Therefore, maximizing the fraction of the proteome that can be captured with high-quality curves will also benefit PISA-type approaches. In the end, optimizing proteome melting or denaturation conditions will allow a compound of interest to be reliably screened against a greater fraction of the cellular proteome and provide a more complete understanding of its mechanism of action.

Overall, the proteome-wide solvent shift assays described here represent powerful tools for unbiased assessments of compound target engagement. By integrating the basic principles SIP with

modern TMTpro-based quantitative proteomics, we have developed a robust assay that is reminiscent of other approaches, like TPP. While we ultimately consider SPP and TPP to be complimentary approaches, we propose that SPP could serve as a suitable alternative for anyone that is unable to perform TPP. Moving forward, we propose that additional strategies focused on maximizing the absolute number of high-quality curves will provide a valuable resource and aid in future drug discovery efforts.

# Materials and methods

## Key resources table

| Reagent type (species) or resource | Designation | Source or reference | Identifiers | Additional information |
|---|---|---|---|---|
| Cell line (*Homo sapiens*) | HCT116 (adult, colorectal cancer) | ATCC | Cat# CCL-247 | Male |
| Antibody | Anti-Akt (rabbit monoclonal) | Cell Signaling Technology | Cat# 4691 | WB (1:1000) |
| Antibody | Anti-p38 MAPK (rabbit polyclonal) | Cell Signaling Technology | Cat# 9212 | WB (1:1000) |
| Antibody | Anti-Aurora A (rabbit monoclonal) | Cell Signaling Technology | Cat# 4718 | WB (1:1000) |
| Antibody | Goat anti-rabbit IgG-HRP | Santa Cruz | Cat# sc-2004 | WB (1:10,000) |
| Chemical compound, drug | SCIO-469 | Cayman Chemical | Cat# 29484; batch: 0575761-1 | 10 µM stock in DMSO |
| Chemical compound, drug | Alisertib (MLN8237) | Cayman Chemical | Cat# 13602; batch: 0565558-18 | 10 µM stock in DMSO |
| Chemical compound, drug | MK2206 (hydrochloride) | Cayman Chemical | Cat# 11593; batch: 0586491-5 | 10 µM stock in DMSO |
| Chemical compound, drug | Vorinostat (SAHA) | Cayman Chemical | Cat# 10009929; batch: 0512249-52 | 10 µM stock in DMSO |
| Commercial assay or kit | TMTpro 16plex Label Reagent Set | Thermo Fisher | Cat# A44520; batch: VI313212 | Solubilized in anhydrous acetonitrile |
| Commercial assay or kit | JNK2 kinase enzyme | Promega | Cat# VA7210 | |
| Commercial assay or kit | ADP-Glo kinase assay | Promega | Cat# V9101 | |
| Software, algorithm | Perseus | ttp://maxquant.net/perseus *Tyanova et al., 2016* | Version 1.6.15.0 | |
| Software, algorithm | Prism | GraphPad | Version 9.0.0 | |
| Software, algorithm | R | https://www.r-project.org/ | Version 4.0.2 | |
| Software, algorithm | TPP package | https://github.com/DoroChilds/TPP; *Franken et al., 2015* | Version 3.17.6 | |
| Other | Sera-Mag SpeedBead Carboxylate-Modified Magnetic Particles (Hydrophobic) | Cytiva | Cat# 44152105050250 | |
| Other | Sera-Mag Carboxylate-Modified Magnetic Particles (Hydrophylic) | Cytiva | Cat# 45152105050250 | |

## Cell culture

HCT116 cells (purchased from ATCC) were cultured in Dulbecco's Modified Eagle's Medium supplemented with 10% fetal bovine serum and 1× penicillin-streptomycin. Cells were grown until ~80% confluency, harvested by scraping, washed with phosphate-buffered saline, and flash frozen in liquid nitrogen. Cell pellets were stored at –80°C until ready for use. HCT116 cells were confirmed to be mycoplasma negative.

## SDS-PAGE and western blotting

HCT116 cell lysates were combined with Laemmli buffer and resolved on Novex WedgeWell 4–20% Tris-Glycine gels (Invitrogen). Gels were either stained with Novex SimplyBlue SafeStain (Invitrogen) or transferred to an Immun-Blot PVDF membrane (Bio-Rad). Membranes were immunoblotted

with antibodies against p38 MAPK (Cell Signaling Technology, 9212), Akt (Cell Signaling Technology, 4691), and Aurora A (Cell Signaling Technology, 4718).

## Solvent proteome profiling

HCT116 proteomes were extracted from frozen cell pellets with lysis buffer (1× TBS, 1.5 mM MgCl$_2$, 0.5% NP-40, 1× protease inhibitor [Pierce Protease Inhibitor Mini Tablets]). Lysates were incubated for 15 min at 4°C and cleared by centrifugation at 20,000 × $g$. The soluble fraction was diluted to 2 mg/mL using lysis buffer and allowed to warm to room temperature (RT). Compound (solubilized in DMSO) or DMSO was added to the desired concentration, and the samples were allowed to incubate at RT for 15 min. Each sample was divided into eight aliquots, each of which was treated with an increasing concentration of AEA from 0 to 21% (0, 3, 6, 9, 12, 15, 18, and 21%). Upon addition of AEA, samples were incubated at 37°C with vigorous shaking for 20 min. Precipitated proteins were removed by centrifugation at 21,000 × $g$ for 15 min. An equal volume of each soluble fraction was collected and prepared for LC-MS/MS analysis.

## Thermal proteome profiling

HCT116 proteomes were extracted from frozen cell pellets with lysis buffer (1× TBS, 1.5 mM MgCl$_2$, 0.5% NP-40, 1× protease inhibitor Pierce Protease Inhibitor Mini Tablets). Lysates were incubated for 15 min at 4°C and cleared by centrifugation at 20,000 × $g$. The soluble fraction was diluted to 2 mg/mL using lysis buffer and allowed to warm to RT. DMSO was added to the desired concentration, and the samples were allowed to incubate at RT for 15 min. Each sample was divided into eight aliquots and heated across a temperature gradient from 38 to 61°C for 3 min in an Eppendorf Mastercycler Pro S. Precipitated proteins were removed by centrifugation at 21,000 × $g$ for 15 min. An equal volume of each soluble fraction was collected and prepared for LC-MS/MS analysis.

## Solvent-PISA assay

HCT116 proteomes were extracted from frozen cell pellets with lysis buffer (1× TBS, 1.5 mM MgCl$_2$, 0.5% NP-40, 1× protease inhibitor [Pierce Protease Inhibitor Mini Tablets]). Lysates were incubated for 15 min at 4°C and cleared by centrifugation at 20,000 × $g$. The soluble fraction was diluted to 2 mg/mL using lysis buffer and allowed to warm to RT. Compound (solubilized in DMSO) or DMSO was added to the desired concentration, and the samples were allowed to incubate at RT for 15 min. Each sample was divided into eight aliquots, each of which was treated with an increasing concentration of AEA from 0 to 21% (0, 3, 6, 9, 12, 15, 18, and 21%), 9 to 19% (9, 10.5, 12, 13.5, 15, 16.5, 18, and 19.5%), 11 to 19% (11, 12.25, 13.5, 14.75, 16, 17.25, 18.5, and 19.75%), or 14 to 19% (14.25, 15, 15.75, 16.5, 17.25, 18, 18.75, and 19.5%). Upon addition of AEA, samples were incubated at 37°C with vigorous shaking for 20 min. Precipitated proteins were removed by centrifugation at 21,000 × $g$ for 15 min. An equal volume of each resulting soluble fraction was pooled into a single sample and prepared for LC-MS/MS analysis.

## Thermal-PISA assay

HCT116 proteomes were extracted from frozen cell pellets with lysis buffer (1× TBS, 1.5 mM MgCl$_2$, 0.5% NP-40, 1× protease inhibitor [Pierce Protease Inhibitor Mini Tablets]). Lysates were incubated for 15 min at 4°C and cleared by centrifugation at 20,000 × $g$. The soluble fraction was diluted to 2 mg/mL using lysis buffer and allowed to warm to RT. Compound (solubilized in DMSO) or DMSO was added to the desired concentration, and the samples were allowed to incubate at RT for 15 min. Each sample was divided into eight aliquots, each of which was heated to a different temperature from 48 to 58°C for 3 min in an Eppendorf Mastercycler Pro S. Samples were allowed to cool at RT for 5 min. An equal volume of each sample was pooled and spun at 21,000 × $g$ for 90 min. Approximately 15 µg of soluble protein were collected and prepared for LC-MS/MS analysis.

## LC-MS/MS sample preparation

Samples (15–20 µg protein) were diluted in prep buffer (400 mM EPPS pH 8.5, 1% SDS, 10 mM tris(2-carboxyethyl)phosphine hydrochloride) and incubated at RT for 10 min. Iodoacetimide was added to a final concentration of 10 mM to each sample and incubated for 25 min in the dark. Finally, DTT was added to each sample to a final concentration of 10 mM. A buffer exchange was carried out using a

modified SP3 protocol (*Hughes et al., 2014*; *Hughes et al., 2019*). Briefly, ~250 µg of each SpeedBead Magnetic Carboxylate modified particles (Cytiva; 45152105050250, 65152105050250) mixed at a 1:1 ratio were added to each sample. 100% ethanol was added to each sample to achieve a final ethanol concentration of at least 50%. Samples were incubated with gentle shaking for 15 min. Samples were washed three times with 80% ethanol. Protein was eluted from SP3 beads using 200 mM EPPS pH 8.5 containing trypsin (Thermo Fisher Scientific) and Lys-C (Wako). Samples were digested overnight at 37°C with vigorous shaking. Acetonitrile was added to each sample to achieve a final concentration of 30%. Each sample was labeled, in the presence of SP3 beads, with ~65 µg of TMTpro 16plex reagents (Thermo Fisher Scientific) (*Li et al., 2020a*; *Thompson et al., 2019*). Experimental layouts for each experiment are described in the corresponding source data tables. Following confirmation of satisfactory labeling (>97%), excess TMTpro reagents were quenched by addition of hydroxylamine to a final concentration of 0.3%. The full volume from each sample was pooled and acetonitrile was removed by vacuum centrifugation for 1 hr. The pooled sample was acidified using formic acid and peptides were desalted using a Sep-Pak Vac 50 mg tC18 cartridge (Waters). Peptides were eluted in 70% acetonitrile, 1% formic acid, and dried by vacuum centrifugation. The peptides were resuspended in 10 mM ammonium bicarbonate pH 8, 5% acetonitrile, and fractionated by basic pH reverse-phase HPLC. In total, 24 fractions were collected. The fractions were dried in a vacuum centrifuge, resuspended in 5% acetonitrile, 1%, formic acid and desalted by stage-tip. Final peptides were eluted in 70% acetonitrile, 1% formic acid, dried, and finally resuspended in 5% acetonitrile, 5% formic acid. In the end, 12 of 24 fractions were analyzed by LC-MS/MS.

## MS data acquisition

Data were collected on an Orbitrap Eclipse mass spectrometer (Thermo Fisher Scientific) coupled to a Proxeon EASY-nLC 1000 LC pump (Thermo Fisher Scientific). Peptides were separated using a 90 min gradient at 500 nL/min on a 30 cm column (i.d. 100 µm, Accucore, 2.6 µm, 150 Å) packed in-house. High-field asymmetric-waveform ion mobility spectroscopy (FAIMS) was enabled during data acquisition with compensation voltages (CVs) set as −40 V, −60 V, and −80 V (*Schweppe et al., 2019*). MS1 data were collected using the Orbitrap (60,000 resolution; maximum injection time 50 ms; AGC $4 \times 10^5$). Determined charge states between 2 and 6 were required for sequencing, and a 60 s dynamic exclusion window was used. Data-dependent mode was set as cycle time (1 s). MS2 scans were performed in the Orbitrap with HCD fragmentation (isolation window 0.5 Da; 50,000 resolution; NCE 36%; maximum injection time 86 ms; AGC $1 \times 10^5$).

## MS data analysis

Raw files were first converted to mzXML, and monoisotopic peaks were reassigned using Monocle (*Rad et al., 2021*). Database searching included all human entries from Uniprot (downloaded in February 2014). The database was concatenated with one composed of all protein sequences in the reversed order. Sequences of common contaminant proteins (e.g., trypsin, keratins, etc.) were appended as well. Searches were performed using the Comet search algorithm. Searches were performed using a 50 ppm precursor ion tolerance and 0.02 Da product ion tolerance. TMTpro on lysine residues and peptide N termini (+304.2071 Da) and carbamidomethylation of cysteine residues (+57.0215 Da) were set as static modifications, while oxidation of methionine residues (+15.9949 Da) was set as a variable modification.

Peptide-spectrum matches (PSMs) were adjusted to a 1% false discovery rate (FDR) (*Elias and Gygi, 2007*). PSM filtering was performed using linear discriminant analysis (LDA) as described previously (*Huttlin et al., 2010*) while considering the following parameters: Comet log expect, different sequence delta Comet log expect (percent difference between the first hit and the next hit with a different peptide sequence), missed cleavages, peptide length, charge state, precursor mass accuracy, and fraction of ions matched. Each run was filtered separately. Protein-level FDR was subsequently estimated at a dataset level. For each protein across all samples, the posterior probabilities reported by the LDA model for each peptide were multiplied to give a protein-level probability estimate. Using the Picked FDR method (*Savitski et al., 2015*), proteins were filtered to the target 1% FDR level.

For reporter ion quantification, a 0.003 Da window around the theoretical *m/z* of each reporter ion was scanned, and the most intense *m/z* was used. Reporter ion intensities were adjusted to correct for the isotopic impurities of the different TMTpro reagents according to the manufacturer's

specifications. Peptides were filtered to include only those with a summed signal-to-noise (SN) of 160 or greater across all channels. For each protein, the filtered peptide TMTpro SN values were summed to generate protein quantification.

## Data analysis for SPP and TPP

The TPP R package (3.17.6) (*Franken et al., 2015*) was adapted to enable the analysis of SPP data. First, a bug in 'analyzeTPPTR.r' was fixed to allow passing starting parameters to the function 'tpptrNormalize.' The starting parameters used for SPP data were 'startPars = c("Pl" = 0,"a" = 55,"b" = 10).' The starting parameters used for TPP data were 'startPars = c("Pl" = 0,"a" = 550,"b" = 10)' (default parameters in TPP package). Second, in 'inflectionPoint.r,' the parameter 'interval' for function 'uniroot' was set as 'c(3,21)' for SPP data, and the temperature range (default parameter in TPP package) for TPP data.

Protein quantifications were first scaled to the first concentration or temperature. The adapted TPP R package was then used to perform the normalization that accounts for technical variance (e.g., pipetting error) (*Savitski et al., 2014*). Proteins for generating normalization curves met the following criteria: (i) the 5th (8-point curves) or 10th (16-point curves) point was between 0.4 and 0.6; (ii) the 7th (8-point curves) or 15th (16-point curves) point was between 0 and 0.3; and (iii) the last point was between 0 and 0.2. Normalized data were then fit to sigmoidal curves using the adapted TPP R package with the parameters described above. Following the criteria used in the published TPP protocol (*Franken et al., 2015*), we designated curves with $R^2 > 0.8$, plateau < 0.3, valid slopes, and valid melting/denaturation points as high-quality curves. Significant hits are designated as proteins fulfilling all four requirements (fulfills_all_4_requirements="Yes", see *Figure 1—source data 1*, *Figure 2—source data 1–3*, *Figure 5—source data 1* for details), with adjusted p-values<0.001 and peptides > 1 in both replicates.

## Data analysis for solvent-PISA and thermal-PISA assays

Solvent-PISA data were analyzed using Perseus (*Tyanova et al., 2016*). Significant changes were determined using a permutation-based FDR with the following settings: FDR – 0.05, S0 – 0.1, and number of randomizations – 250. Only proteins that were quantified with >1 peptide were analyzed. Individual fold change values were calculated in reference to the mean of the vehicle-treated samples.

For solvent- and thermal-PISA datasets, proteins quantified with <2 peptides were removed. For comparison of solvent- and thermal-PISA, a %CV was calculated for the treated and vehicle groups. Proteins in which either or both %CV values were >15% were removed. The remaining data were ranked by log2 fold change.

Theoretical fold changes in solvent-PISA were calculated from respective SPP data for *Figure 4G*. The normalized protein abundance in the output table of TPP package was used. The normalized protein abundance was summed for the compound-treated samples or vehicle-treated controls at each %AEA within a given range. Considering the 0–21% window (window 1), for example, we simply summed all eight abundance measurements from the eight AEA concentrations. For the 9–19% window (window 2), we summed only the protein abundances between 9 and 19%, and so on. Then the fold changes were calculated as summed protein abundance for compound-treated samples versus that for vehicle-treated samples.

## In vitro kinase assay

MAPK9/JNK2 in vitro kinase assay was performed using the JNK2 kinase assay kit (Promega #VA7210) and the ADP-Glo kit (Promega #V9101) according to the manufacturer's instructions. In brief, SCIO-469 or 5% DMSO (1 µL) was added to recombinant MAPK9/JNK2 (10 or 20 ng in 2 µL) in a white 384-well assay plate (Corning #3825). To this was added 2 µL of a solution containing the peptide substrate and ATP to a final, in-well concentration of 0.2 µg/µL and 50 µM, respectively. The kinase reaction proceeded at RT (1 hr) before adding ADP-Glo reagent (5 µL), which was incubated at RT (45 min). Finally, the kinase detection reagent was added (10 µL) and the luminescence was measured after 1 hr using an EnVision plate reader (PerkinElmer). Background luminescent signal, measured from no enzyme controls, was subtracted from each measurement before calculating relative percent inhibition. All experimental conditions were performed in quadruplicate. Plots and statistics were generated using GraphPad Prism (v7).

## Gene Ontology enrichment analysis

Gene Ontology enrichment analysis was performed with DAVID (6.8) (*Huang et al., 2009*). All quantified proteins in the respective dataset were used as the background. p-Values were corrected with Benjamini–Hochberg method and filtered at 0.001 unless otherwise stated.

## Acknowledgements

We would like to thank the members of the Gygi lab at Harvard Medical School. This work was supported in part by a grant from the NIH (GM097645) to SPG, a fellowship. JGV is the Mark Foundation for Cancer Research Fellow of the Damon Runyon Cancer Research Foundation (DRG-2359-19).

## Additional information

### Funding

| Funder | Grant reference number | Author |
| --- | --- | --- |
| Damon Runyon Cancer Research Foundation | DRG-2359-19 | Jonathan G Van Vranken |
| Mark Foundation For Cancer Research | DRG-2359-19 | Jonathan G Van Vranken |
| National Institutes of Health | GM097645 | Steven P Gygi |

The funders had no role in study design, data collection and interpretation, or the decision to submit the work for publication.

### Author contributions

Jonathan G Van Vranken, Conceptualization, Data curation, Formal analysis, Funding acquisition, Investigation, Methodology, Writing - original draft, Writing – review and editing; Jiaming Li, Steven P Gygi, Conceptualization, Data curation, Formal analysis, Funding acquisition, Investigation, Methodology, Software, Writing – review and editing; Dylan C Mitchell, Investigation; José Navarrete-Perea, Formal analysis, Instrumentation, Software, Writing – review and editing

### Author ORCIDs

Jonathan G Van Vranken ⓘ http://orcid.org/0000-0002-8931-852X
Jiaming Li ⓘ http://orcid.org/0000-0002-9065-6913
José Navarrete-Perea ⓘ http://orcid.org/0000-0001-6265-6214
Steven P Gygi ⓘ http://orcid.org/0000-0001-7626-0034

### Decision letter and Author response

Decision letter https://doi.org/10.7554/eLife.70784.sa1
Author response https://doi.org/10.7554/eLife.70784.sa2

## Additional files

### Supplementary files

• Transparent reporting form

### Data availability

MS raw files and search files have been uploaded to PRIDE with dataset identifier PXD026297.

The following dataset was generated:

| Author(s) | Year | Dataset title | Dataset URL | Database and Identifier |
| --- | --- | --- | --- | --- |
| Van Vranken JG | 2021 | Assessing target engagement using proteome-wide solvent shift assays | http://proteomecentral.proteomexchange.org/cgi/GetDataset?ID=PXD026297 | ProteomeXchange, PXD026297 |

The following previously published datasets were used:

| Author(s) | Year | Dataset title | Dataset URL | Database and Identifier |
| --- | --- | --- | --- | --- |
| Jarzab A | 2020 | Meltome atlas-thermal proteome stability across the tree of life | http://proteomecentral.proteomexchange.org/cgi/GetDataset?ID=PXD011929 | ProteomeXchange, PXD011929 |

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
