## [Editor Report]

This manuscript will be of broad interest to readers in the field of proteomics and drug discovery. It describes a potentially robust method for the identification of biological targets of small molecules, a substantial hurdle in drug discovery. The experiments described are rigorous, and this manuscript provides a useful template for the broad implementation of this method.

---

## [Decision Letter]

**Decision letter after peer review:**

Thank you for submitting your article "Assessing target engagement using proteome-wide solvent shift assays" for consideration by *eLife*. Your article has been reviewed by 3 peer reviewers, and the evaluation has been overseen by a Reviewing Editor and Michael Marletta as the Senior Editor. The reviewers have opted to remain anonymous.

Essential revisions:

1) Carryout experiments to validate one of the new small molecule – protein targets identified by SPP or solvent-PISA, using independent method or approach. For example, biochemical validation (i.e. direct small molecule-protein binding or in vitro activity assays) for either of these discoveries (SCIO-469 – MAPK9, Vorinostat – HDAC6, MK2206 – CHEK1). This would help to improve the novelty of the work and is the consensus of reviewers and reviewing editor.

2) Revise the manuscript according to Reviewer 2 and 3's comments. This does not require new experiments.

*Reviewer #1 (Recommendations for the authors):*

To improve this study, it would be great if the authors could validate one of the new small molecule – protein targets they identified by SPP or solvent-PISA, using independent method or approach. For example, biochemical validation (i.e. direct small molecule-protein binding or in vitro activity assays) for either of these discoveries would be terrific (SCIO-469 – MAPK9, Vorinostat – HDAC6, MK2206 – CHEK1).

*Reviewer #2 (Recommendations for the authors):*

The authors note interesting observations for during their profiling – notably, the apparent destabilization of certain targets. For example, many targets (e.g. DLD) upon treatment with vorinstat (figure 3b), a few targets for alisertib (figure 4, supp1, B) and CHEK1 for MK2206 (Figure 4, supp 2, B).

In some cases, it appears that similar numbers of proteins are significantly de-stabilized as stabilized, specifically in the case of vorinostat. Is this artifactual or do the authors believe these changes to be reflective biological changes? Considering these experiments are performed in cell lysates, this raises doubts that such changes are the result of inhibitor-induced biological changes.

The authors should provide an explanation for these observations.

*Reviewer #3 (Recommendations for the authors):*

A plot demonstrating the correlation (or absence thereof) between the Cm and Tm of unperturbed lysates would help a lot, as well as a plot between δ(Tm) and δ(Cm) for treated lysates.

---

## [Author Response]

Essential revisions:1) Carryout experiments to validate one of the new small molecule – protein targets identified by SPP or solvent-PISA, using independent method or approach. For example, biochemical validation (i.e. direct small molecule-protein binding or in vitro activity assays) for either of these discoveries (SCIO-469 – MAPK9, Vorinostat – HDAC6, MK2206 – CHEK1). This would help to improve the novelty of the work and is the consensus of reviewers and reviewing editor.

This is an excellent suggestion from the reviewers. We chose to validate the engagement of SCIO-469 by MAPK9/JNK2. To do so, we performed an in vitro kinase assay in which recombinant MAPK9/JNK2 was exposed to 10 µM SCIO-469. We found that SCIO-469 significantly inhibited the activity of MAPK9/JNK2. This data can be found in Figure 3 —figure supplement 1B.

2) Revise the manuscript according to Reviewer 2 and 3's comments. This does not require new experiments.

The revisions we made in response to Reviewer 2 and 3’s comments are summarized below.

Reviewer #1 (Recommendations for the authors):To improve this study, it would be great if the authors could validate one of the new small molecule – protein targets they identified by SPP or solvent-PISA, using independent method or approach. For example, biochemical validation (i.e. direct small molecule-protein binding or in vitro activity assays) for either of these discoveries would be terrific (SCIO-469 – MAPK9, Vorinostat – HDAC6, MK2206 – CHEK1).

We thank the reviewer for this suggestion. We validated the engagement of SCIO-469 by MAPK9/JNK2 using an in vitro kinase assay, in which recombinant MAPK9/JNK2 was exposed to 10 µM SCIO-469. We found that SCIO-469 significantly inhibited the activity of MAPK9/JNK2. This data can be found in Figure 3 —figure supplement 1B.

Reviewer #2 (Recommendations for the authors):The authors note interesting observations for during their profiling – notably, the apparent destabilization of certain targets. For example, many targets (e.g. DLD) upon treatment with vorinstat (figure 3b), a few targets for alisertib (figure 4, supp1, B) and CHEK1 for MK2206 (Figure 4, supp 2, B).In some cases, it appears that similar numbers of proteins are significantly de-stabilized as stabilized, specifically in the case of vorinostat. Is this artifactual or do the authors believe these changes to be reflective biological changes? Considering these experiments are performed in cell lysates, this raises doubts that such changes are the result of inhibitor-induced biological changes.The authors should provide an explanation for these observations.

This is a very inciteful observation and something that we have discussed in the past. Generally speaking, we think about destabilization the same as stabilization. That is, a thermal- or chemical-destabilization can provide evidence of target engagement. Vorinostat is a unique case as there appear to be dozens of destabilized proteins. In our experience with this assay, we have only ever observed this effect with vorinostat. We still don’t know the exact cause. However, we do not believe that the destabilization of these proteins is the result of direct engagement but, rather, has something to do with the chemical properties of vorinostat, specifically. Importantly, we see the exact same effect with thermal-PISA (see Author response image 1). Therefore, this effect seems to be unique to vorinostat and not a deficiency of SPP.

**Author response image 1. sa2fig1:** HCT116 lysates were treated with 25 µM vorinostat and assays by thermal-PISA (N=2). We included this explanation in the Results section of the main text (page 10).

Reviewer #3 (Recommendations for the authors):A plot demonstrating the correlation (or absence thereof) between the Cm and Tm of unperturbed lysates would help a lot, as well as a plot between δ(Tm) and δ(Cm) for treated lysates.

We thank the reviewer for this suggestion and hope that including these plots will be helpful for readers. We have generated a plot comparing T_M_ and C_M_ and included this in Figure 5 —figure supplement 1B. The manuscript is about solvent proteome profiling and solvent-PISA, and we did not collect any thermal proteome profiling data in the context of drug treatments. So we are unable to generate a plot comparing deltaT_M_ and deltaC_M_ for any compounds.